# Time–Data Tradeoffs by Aggressive Smoothing

**John J. Bruer[1,*]**     **Joel A. Tropp[1]**     **Volkan Cevher[2]**     **Stephen R. Becker[3]**

[1]Dept. of Computing + Mathematical Sciences, California Institute of Technology
[2]Laboratory for Information and Inference Systems, EPFL
[3]Dept. of Applied Mathematics, University of Colorado at Boulder
[*]`jbruer@cms.caltech.edu`

## Abstract

This paper proposes a tradeoff between sample complexity and computation time that applies to statistical estimators based on convex optimization. As the amount of data increases, we can smooth optimization problems more and more aggressively to achieve accurate estimates more quickly. This work provides theoretical and experimental evidence of this tradeoff for a class of regularized linear inverse problems.

## 1   Introduction

It once seemed obvious that the running time of an algorithm should increase with the size of the input. But recent work in machine learning has led us to question this dogma. In particular, Shalev-Shwartz and Srebro [1] showed that their algorithm for learning a support vector classifier actually becomes *faster* when they increase the amount of training data. Other researchers have identified related tradeoffs [2, 3, 4, 5, 6, 7, 8, 9]. Together, these works support an emerging perspective in statistical computation that treats data as a computational resource that we can exploit to improve algorithms for estimation and learning.

In this paper, we consider statistical algorithms based on convex optimization. Our primary contribution is the following proposal:

> *As the amount of available data increases, we can smooth statistical optimization problems more and more aggressively. We can solve the smoothed problems significantly faster without any increase in statistical risk.*

Indeed, many statistical estimation procedures balance the modeling error with the complexity of the model. When we have very little data, complexity regularization is essential to fit an accurate model. When we have a large amount of data, we can relax the regularization without compromising the quality of the model. In other words, excess data offers us an opportunity to accelerate the statistical optimization. We propose to use smoothing methods [10, 11, 12] to implement this tradeoff.

We develop this idea in the context of the *regularized linear inverse problem* (RLIP) with random data. Nevertheless, our ideas apply to a wide range of problems. We pursue a more sophisticated example in a longer version of this work [13].

JJB's and JAT's work was supported under ONR award N00014-11-1002, AFOSR award FA9550-09-1-0643, and a Sloan Research Fellowship. VC's work was supported in part by the European Commission under Grant MIRG-268398, ERC Future Proof, SNF 200021-132548, SNF 200021-146750 and SNF CRSII2-147633. SRB was previously with IBM Research, Yorktown Heights, NY 10598 during the completion of this work.

## 1.1 The regularized linear inverse problem

Let $x^\natural \in \mathbb{R}^d$ be an unknown signal, and let $A \in \mathbb{R}^{m \times d}$ be a known measurement matrix. Assume that we have access to a vector $b \in \mathbb{R}^m$ of $m$ linear samples of that signal given by

$$b := Ax^\natural.$$

Given the pair $(A, b)$, we wish to recover the original signal $x^\natural$.

We consider the case where $A$ is fat ($m < d$), so we cannot recover $x^\natural$ without additional information about its structure. Let us introduce a proper convex function $f \colon \mathbb{R}^d \to \mathbb{R} \cup \{+\infty\}$ that assigns small values to highly structured signals. Using the regularizer $f$, we construct the estimator

$$\widehat{x} := \arg\min_x f(x) \quad \text{subject to} \quad Ax = b. \tag{1}$$

We declare the estimator successful when $\widehat{x} = x^\natural$, and we refer to this outcome as *exact recovery*.

While others have studied (1) in the statistical setting, our result is different in character from previous work. Agarwal, Negahban, and Wainwright [14] showed that gradient methods applied to problems like (1) converge in fewer iterations due to increasing restricted strong convexity and restricted smoothness as sample size increases. They did not, however, discuss a time–data tradeoff explicitly, nor did they recognize that the overall computational cost may rise as the problem sizes grow.

Lai and Yin [15], meanwhile, proposed relaxing the regularizer in (1) based solely on some norm of the underlying signal. Our relaxation, however, is based on the *sample size* as well. Our method results in better performance as sample size increases: a time–data tradeoff.

The RLIP (1) provides a good candidate for studying time–data tradeoffs because recent work in convex geometry [16] gives a precise characterization of the number of samples needed for exact recovery. Excess samples allow us to replace the optimization problem (1) with one that we can solve faster. We do this for sparse vector and low-rank matrix recovery problems in Sections 4 and 5.

## 2 The geometry of the time–data tradeoff

In this section, we summarize the relevant results that describe the minimum sample size required to solve the regularized linear inverse problem (1) exactly in a statistical setting.

### 2.1 The exact recovery condition and statistical dimension

We can state the optimality condition for (1) in a geometric form; cf. [17, Prop. 2.1].

**Fact 2.1** (Exact recovery condition). *The descent cone of a proper convex function $f \colon \mathbb{R}^d \to \mathbb{R} \cup \{+\infty\}$ at the point $x$ is the convex cone*

$$\mathcal{D}(f; x) := \bigcup_{\tau > 0} \left\{ y \in \mathbb{R}^d : f(x + \tau y) \leq f(x) \right\}.$$

*The regularized linear inverse problem* (1) exactly recovers *the unknown signal $x^\natural$ if and only if*

$$\mathcal{D}(f; x^\natural) \cap \mathrm{null}(A) = \{0\}. \tag{2}$$

We illustrate this condition in Figure 1(a).

To determine the number of samples we need to ensure that the exact recovery condition (2) holds, we must quantify the "size" of the descent cones of the regularizer $f$.

**Definition 2.2** (Statistical dimension [16, Def. 2.1]). *Let $C \in \mathbb{R}^d$ be a convex cone. Its statistical dimension $\delta(C)$ is defined as*

$$\delta(C) := \mathbb{E}\left[ \|\mathbf{\Pi}_C(g)\|^2 \right],$$

where $g \in \mathbb{R}^d$ has independent standard Gaussian entries, and $\mathbf{\Pi}_C$ is the projection operator onto $C$.

When the measurement matrix $A$ is sufficiently random, Amelunxen et al. [16] obtain a precise characterization of the number $m$ of samples required to achieve exact recovery.

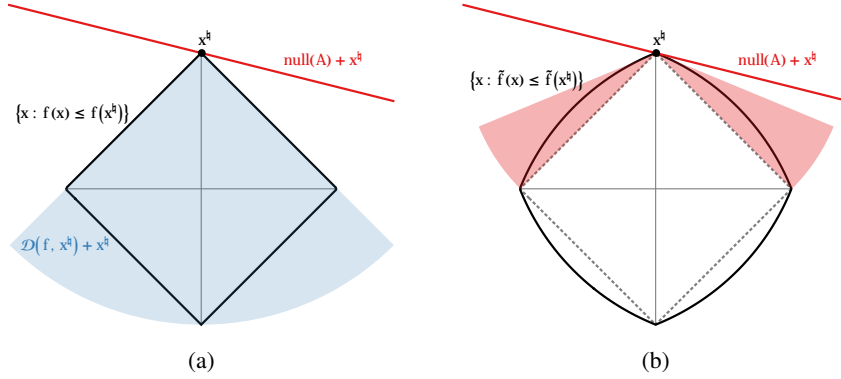

(a)                                      (b)

Figure 1: **The geometric opportunity.** Panel (a) illustrates the exact recovery condition (2). Panel (b) shows a relaxed regularizer $\tilde{f}$ with larger sublevel sets. The shaded area indicates the difference between the descent cones of $\tilde{f}$ and $f$ at $\boldsymbol{x}^{\natural}$. When we have excess samples, Fact 2.3 tells us that the exact recovery condition holds with high probability, as in panel (a). A suitable relaxtion will maintain exact recovery, as in panel (b), while allowing us to solve the problem faster.

**Fact 2.3** (Exact recovery condition for the random RLIP [16, Thm. II])**.** *Assume that the null space of the measurement matrix $\boldsymbol{A} \in \mathbb{R}^{m \times d}$ in the RLIP* (1) *is oriented uniformly at random. (In particular, a matrix with independent standard Gaussian entries has this property.) Then*

$$m \geq \delta\left(\mathcal{D}(f; \boldsymbol{x}^{\natural})\right) + C_{\eta}\sqrt{d} \implies \text{exact recovery holds with probability} \geq 1 - \eta;$$

$$m \leq \delta\left(\mathcal{D}(f; \boldsymbol{x}^{\natural})\right) - C_{\eta}\sqrt{d} \implies \text{exact recovery holds with probability} \leq \eta,$$

*where $C_{\eta} := \sqrt{8\log(4/\eta)}$.*

In words, the RLIP undergoes a phase transition when the number $m$ of samples equals $\delta(\mathcal{D}(f; \boldsymbol{x}^{\natural}))$. Any additional samples are redundant, so we can try to exploit them to identify $\boldsymbol{x}^{\natural}$ more quickly.

### 2.2 A geometric opportunity

Chandrasekaran and Jordan [6] have identified a time–data tradeoff in the setting of denoising problems based on Euclidean projection onto a constraint set. They argue that, when they have a large number of samples, it is possible to enlarge the constraint set without increasing the statistical risk of the estimator. They propose to use a discrete sequence of relaxations based on algebraic hierarchies.

We have identified a related opportunity for a time–data tradeoff in the RLIP (1). When we have excess samples, we may replace the regularizer $f$ with a relaxed regularizer $\tilde{f}$ that is easier to optimize. In contrast to [6], we propose to use a continuous sequence of relaxations based on smoothing.

Figure 1 illustrates the geometry of our time–data tradeoff. When the number of samples exceeds $\delta(\mathcal{D}(f; \boldsymbol{x}^{\natural}))$, Fact 2.3 tells us that the situation shown in Figure 1(a) holds with high probability. This allows us to enlarge the sublevel sets of the regularizer while still satisfying the exact recovery condition, as shown in Figure 1(b). A suitable relaxation allows us to solve the problem faster. Our geometric motivation is similar with [6] although our relaxation method is totally unrelated.

## 3 A time–data tradeoff via dual-smoothing

This section presents an algorithm that can exploit excess samples to solve the RLIP (1) faster.

### 3.1 The dual-smoothing procedure

The procedure we use applies Nesterov's primal-smoothing method from [11] to the dual problem; see [12]. Given a regularizer $f$, we introduce a family $\{f_{\mu} : \mu > 0\}$ of strongly convex majorants:

$$f_{\mu}(\boldsymbol{x}) := f(\boldsymbol{x}) + \frac{\mu}{2}\|\boldsymbol{x}\|^2.$$

**Algorithm 3.1** Auslender–Teboulle applied to the dual-smoothed RLIP

---
**Input:** measurement matrix $\boldsymbol{A}$, observed vector $\boldsymbol{b}$
1: $\boldsymbol{z}_0 \leftarrow \boldsymbol{0}, \bar{\boldsymbol{z}}_0 \leftarrow \boldsymbol{z}_0, \theta_0 \leftarrow 1$
2: **for** $k = 0, 1, 2, \ldots$ **do**
3:      $\boldsymbol{y}_k \leftarrow (1 - \theta_k)\boldsymbol{z}_k + \theta_k \bar{\boldsymbol{z}}_k$
4:      $\boldsymbol{x}_k \leftarrow \arg\min_{\boldsymbol{x}} f(\boldsymbol{x}) + \frac{\mu}{2} \|\boldsymbol{x}\|^2 - \langle \boldsymbol{y}_k, \boldsymbol{A}\boldsymbol{x} - \boldsymbol{b} \rangle$
5:      $\bar{\boldsymbol{z}}_{k+1} \leftarrow \bar{\boldsymbol{z}}_k + \frac{\mu}{\|\boldsymbol{A}\|^2 \theta}(\boldsymbol{b} - \boldsymbol{A}\boldsymbol{x}_k)$
6:      $\boldsymbol{z}_{k+1} \leftarrow (1 - \theta_k)\boldsymbol{z}_k + \theta_k \bar{\boldsymbol{z}}_{k+1}$
7:      $\theta_{k+1} \leftarrow 2/(1 + (1 + 4/\theta_k^2)^{1/2})$
8: **end for**

---

In particular, the sublevel sets of $f_\mu$ grow as $\mu$ increases. We then replace $f$ with $f_\mu$ in the original RLIP (1) to obtain new estimators of the form

$$\widehat{\boldsymbol{x}}_\mu := \arg\min_{\boldsymbol{x}} f_\mu(\boldsymbol{x}) \quad \text{subject to} \quad \boldsymbol{A}\boldsymbol{x} = \boldsymbol{b}. \tag{3}$$

The Lagrangian of the convex optimization problem (3) becomes

$$\mathcal{L}_\mu(\boldsymbol{x}, \boldsymbol{z}) = f(\boldsymbol{x}) + \frac{\mu}{2} \|\boldsymbol{x}\|^2 - \langle \boldsymbol{z}, \boldsymbol{A}\boldsymbol{x} - \boldsymbol{b} \rangle,$$

where the Lagrange multiplier $\boldsymbol{z}$ is a vector in $\mathbb{R}^m$. This gives a family of dual problems:

$$\text{maximize} \quad g_\mu(\boldsymbol{z}) := \min_{\boldsymbol{x}} \mathcal{L}_\mu(\boldsymbol{x}, \boldsymbol{z}) \quad \text{subject to} \quad \boldsymbol{z} \in \mathbb{R}^m. \tag{4}$$

Since $f_\mu$ is strongly convex, the Lagrangian $\mathcal{L}$ has a unique minimizer $\boldsymbol{x}_{\boldsymbol{z}}$ for each dual point $\boldsymbol{z}$:

$$\boldsymbol{x}_{\boldsymbol{z}} := \arg\min_{\boldsymbol{x}} \mathcal{L}_\mu(\boldsymbol{x}, \boldsymbol{z}). \tag{5}$$

Strong duality holds for (3) and (4) by Slater's condition [18, Sec. 5.2.3]. Therefore, if we solve the dual problem (4) to obtain an optimal dual point, (5) returns the unique optimal primal point.

The dual function is differentiable with $\nabla g_\mu(\boldsymbol{z}) = \boldsymbol{b} - \boldsymbol{A}\boldsymbol{x}_{\boldsymbol{z}}$, and the gradient is Lipschitz-continuous with Lipschitz constant $L_\mu$ no larger than $\mu^{-1} \|\boldsymbol{A}\|^2$; see [12, 11]. Note that $L_\mu$ is decreasing in $\mu$, and so we call $\mu$ the *smoothing parameter*.

## 3.2 Solving the smoothed dual problem

In order to solve the smoothed dual problem (4), we apply the fast gradient method from Auslender and Teboulle [19]. We present the pseudocode in Algorithm 3.1.

The computational cost of the algorithm depends on two things: the number of iterations necessary for convergence and the cost of each iteration. The following result bounds the error of the primal iterates $\boldsymbol{x}_k$ with respect to the true signal $\boldsymbol{x}^\natural$. The proof is in the supplemental material.

**Proposition 3.1** (Primal convergence of Algorithm 3.1). *Assume that the exact recovery condition holds for the primal problem* (3). *Algorithm 3.1 applied to the smoothed dual problem* (4) *converges to an optimal dual point* $\boldsymbol{z}_\mu^\star$. *Let* $\boldsymbol{x}_\mu^\star$ *be the corresponding optimal primal point given by* (5). *Then the sequence of primal iterates* $\{\boldsymbol{x}_k\}$ *satisfies*

$$\|\boldsymbol{x}^\natural - \boldsymbol{x}_k\| \leq \frac{2\|\boldsymbol{A}\|\|\boldsymbol{z}_\mu^\star\|}{\mu \cdot k}.$$

The chosen regularizer affects the cost of Algorithm 3.1, line 4. Fortunately, this step is inexpensive for many regularizers of interest. Since the matrix–vector product $\boldsymbol{A}\boldsymbol{x}_k$ in line 5 dominates the other vector arithmetic, each iteration requires $O(md)$ arithmetic operations.

## 3.3 The time–data tradeoff

Proposition 3.1 suggests that increasing the smoothing parameter $\mu$ leads to faster convergence of the primal iterates of the Auslender–Teboulle algorithm. The discussion in Section 2.2 suggests that, when we have excess samples, we can increase the smoothing parameter while maintaining exact recovery. Our main technical proposal combines these two observations:

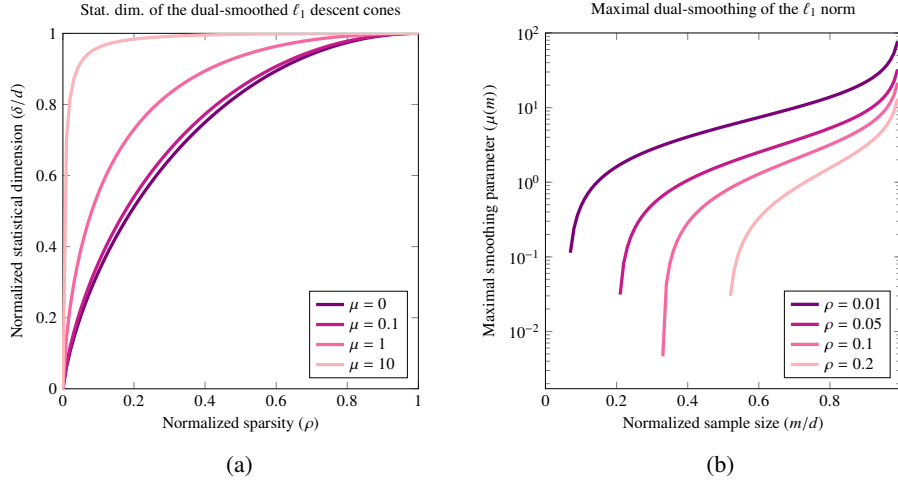

Figure 2: **Statistical dimension and maximal smoothing for the dual-smoothed $\ell_1$ norm.**
Panel (a) shows upper bounds for the normalized statistical dimension $d^{-1}\mathcal{D}(f_\mu; x^\natural)$ of the dual-smoothed sparse vector recovery problem for several choices of $\mu$. Panel (b) shows lower bounds for the maximal smoothing parameter $\mu(m)$ for several choices of the normalized sparsity $\rho := s/d$.

> *As the number m of measurements in the RLIP* (1) *increases, we smooth the dual problem* (4) *more and* more *aggressively while maintaining exact recovery. The Auslender–Teboulle algorithm can solve these increasingly smoothed problems faster.*

In order to balance the inherent tradeoff between smoothing and accuracy, we introduce the *maximal smoothing parameter* $\mu(m)$. For a sample size $m$, $\mu(m)$ is the largest number satisfying

$$\delta\left(\mathcal{D}(f_{\mu(m)}; x^\natural)\right) \leq m. \tag{6}$$

Choosing a smoothing parameter $\mu \leq \mu(m)$ ensures that we do not cross the phase transition of our RLIP. In practice, we need to be less aggressive in order to avoid the "transition region". The following two sections provide examples that use our proposal to achieve a clear time–data tradeoff.

## 4 Example: Sparse vector recovery

In this section, we apply the method outlined in Section 3 to the sparse vector recovery problem.

### 4.1 The optimization problem

Assume that $x^\natural$ is a sparse vector. The $\ell_1$ norm serves as a convex proxy for sparsity, so we choose it as the regularizer in the RLIP (1). This problem is known as *basis pursuit*, and it was proposed by Chen et al. [20]. It has roots in geophysics [21, 22].

We apply the dual-smoothing procedure from Section 3 to obtain the relaxed primal problem, which is equivalent to the *elastic net* of Zou and Hastie [23]. The smoothed dual is given by (4).

To determine the exact recovery condition, Fact 2.3, for the dual-smoothed RLIP (3), we must compute the statistical dimension of the descent cones of $f_\mu$. We provide an accurate upper bound.

**Proposition 4.1** (Statistical dimension bound for the dual-smoothed $\ell_1$ norm). *Let $x \in \mathbb{R}^d$ with $s$ nonzero entries, and define the normalized sparsity $\rho := s/d$. Then*

$$\frac{1}{d}\delta\left(\mathcal{D}(f_\mu; x)\right) \leq \inf_{\tau \geq 0}\left\{\rho\left[1 + \tau^2(1 + \mu\|x\|_{\ell_\infty})^2\right] + (1-\rho)\sqrt{\frac{2}{\pi}}\int_\tau^\infty (u-\tau)^2 e^{-u^2/2}\,\mathrm{d}u\right\}.$$

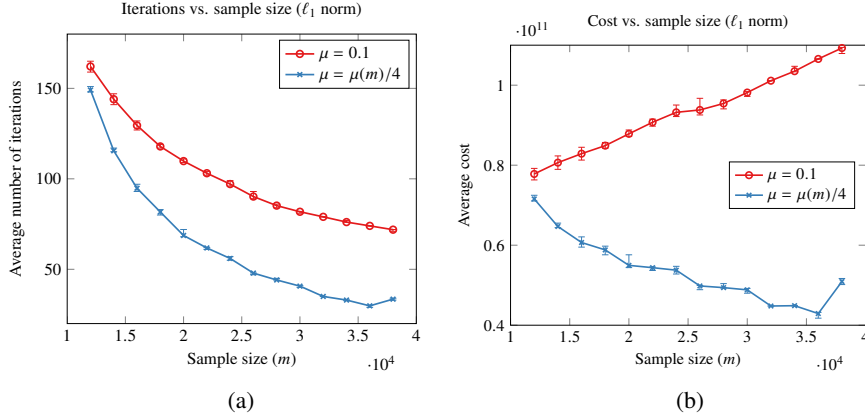

Figure 3: **Sparse vector recovery experiment.** The average number of iterations (a) and the average computational cost (b) of 10 random trials of the dual-smoothed sparse vector recovery problem with ambient dimension $d = 40\,000$ and normalized sparsity $\rho = 5\%$ for various sample sizes $m$. The red curve represents a fixed smoothing parameter $\mu = 0.1$, while the blue curve uses $\mu = \mu(m)/4$. The error bars indicate the minimum and maximum observed values.

The proof is provided in the supplemental material. Figure 2 shows the statistical dimension and maximal smoothing curves for sparse vectors with $\pm 1$ entries. In order to apply this result we only need estimates of the magnitude and sparsity of the signal.

To apply Algorithm 3.1 to this problem, we must calculate an approximate primal solution $x_z$ from a dual point $z$ (Algorithm 3.1, line 4). This step can be written as

$$x_z \leftarrow \mu(m)^{-1} \cdot \text{SoftThreshold}(A^T z, 1),$$

where $[\text{SoftThreshold}(x,t)]_i = \text{sgn}(x_i) \cdot \max\{|x_i| - t, 0\}$. Algorithm 3.1, line 5 dominates the total cost of each iteration.

## 4.2 The time–data tradeoff

We can obtain theoretical support for the existence of a time–data tradeoff in the sparse recovery problem by adapting Proposition 3.1. See the supplemental material for the proof.

**Proposition 4.2** (Error bound for dual-smoothed sparse vector recovery). *Let $x^\natural \in \mathbb{R}^d$ with $s$ nonzero entries, $m$ be the sample size, and $\mu(m)$ be the maximal smoothing parameter* (6). *Given a measurement matrix $A \in \mathbb{R}^{m \times d}$, assume the exact recovery condition* (2) *holds for the dual-smoothed sparse vector recovery problem. Then the sequence of primal iterates from Algorithm 3.1 satisfies*

$$\|x^\natural - x_k\| \le \frac{2 d^{\frac{1}{2}} \kappa(A) \left[ \rho \cdot (1 + \mu(m) \|x^\natural\|_{\ell_\infty})^2 + (1 - \rho) \right]^{\frac{1}{2}}}{\mu(m) \cdot k},$$

*where $\rho := s/d$ is the normalized sparsity of $x^\natural$, and $\kappa(A)$ is the condition number of the matrix $A$.*

For a fixed number $k$ of iterations, as the number $m$ of samples increases, Proposition 4.2 suggests that the error decreases like $1/\mu(m)$. This observation suggests that we can achieve a time–data tradeoff by smoothing.

## 4.3 Numerical experiment

Figure 3 shows the results of a numerical experiment that compares the performance difference between current numerical practice and our aggressive smoothing approach.

Most practitioners use a fixed smoothing parameter $\mu$ that depends on the ambient dimension or sparsity but *not* on the sample size. For the constant smoothing case, we choose $\mu = 0.1$ based on the recommendation in [15]. It is common, however, to see much smaller choices of $\mu$ [24, 25].

In contrast, our method exploits excess samples by smoothing the dual problem more aggressively. We set the smoothing parameter $\mu = \mu(m)/4$. This heuristic choice is small enough to avoid the phase transition of the RLIP while large enough to reap performance benefits. Our forthcoming work [13] addressing the case of noisy samples provides a more principled way to select this parameter.

In the experiment, we fix both the ambient dimension $d = 40\,000$ and the normalized sparsity $\rho = 5\%$. To test each smoothing approach, we generate and solve 10 random sparse vector recovery models for each value of the sample size $m = 12\,000, 14\,000, 16\,000, \ldots, 38\,000$. Each random model comprises a Gaussian measurement matrix $A$ and a random sparse vector $x^{\natural}$ whose nonzero entires are $\pm 1$ with equal probability. We stop Algorithm 3.1 when the relative error $\|x^{\natural} - x_k\| / \|x^{\natural}\|$ is less than $10^{-3}$. This condition guarantees that both methods maintain the same level of accuracy.

In Figure 3(a), we see that for both choices of $\mu$, the average number of iterations decreases as sample size increases. When we plot the total computational cost[1] in Figure 3(b), we see that the constant smoothing method cannot overcome the increase in cost per iteration. In fact, in this example, it would be better to throw away excess data when using constant smoothing. Meanwhile, our aggressive smoothing method manages to *decrease* total cost as sample size increases. The maximal speedup achieved is roughly 2.5×.

We note that if the matrix $A$ were orthonormal, the cost of both smoothing methods would decrease as sample sizes increase. In particular, the uptick seen at $m = 38\,000$ in Figure 3 would disappear (but our method would maintain roughly the same relative advantage over constant smoothing). This suggests that the condition number $\kappa(A)$ indeed plays an important role in determining the computational cost. We believe that using a Gaussian matrix $A$ is warranted here as statistical models often use independent subjects.

Let us emphasize that we use the same algorithm to test both smoothing approaches, so the relative comparison between them is meaningful. The observed improvement shows that we have indeed achieved a time–data tradeoff by aggressive smoothing.

# 5   Example: Low-rank matrix recovery

In this section, we apply the method outlined in Section 3 to the low-rank matrix recovery problem.

## 5.1   The optimization problem

Assume that $X^{\natural} \in \mathbb{R}^{d_1 \times d_2}$ is low-rank. Consider a known measurement matrix $A \in \mathbb{R}^{m \times d}$, where $d := d_1 d_2$. We are given linear measurements of the form $b = A \cdot \mathrm{vec}(X^{\natural})$, where vec returns the (column) vector obtained by stacking the columns of the input matrix. Fazel [26] proposed using the Schatten 1-norm $\|\cdot\|_{S_1}$, the sum of the matrix's singular values, as a convex proxy for rank. Therefore, we follow Recht et al. [27] and select $f = \|\cdot\|_{S_1}$ as the regularizer in the RLIP (1). The low-rank matrix recovery problem has roots in control theory [28].

We apply the dual-smoothing procedure to obtain the approximate primal problem and the smoothed dual problem, replacing the squared Euclidean norm in (3) with the squared Frobenius norm.

As in the sparse vector case, we must compute the statistical dimension of the descent cones of the strongly convex regularizer $f_\mu$. In the case where the matrix $X$ is square, the following is an accurate upper bound for this quantity. (The non-square case is addressed in the supplemental material.)

**Proposition 5.1** (Statistical dimension bound for the dual-smoothed Schatten 1-norm)**.** *Let* $X \in \mathbb{R}^{d_1 \times d_1}$ *have rank* $r$, *and define the normalized rank* $\rho := r/d_1$. *Then*

$$\frac{1}{d_1^2} \delta\left(\mathcal{D}(f_\mu; X)\right) \le \inf_{0 \le \tau \le 2} \left\{ \rho + (1 - \rho) \left[ \rho \left(1 + \tau^2(1 + \mu \|X\|)^2\right) \right.\right.$$

$$\left.\left. + \frac{(1-\rho)}{12\pi} \left[ 24(1 + \tau^2) \cos^{-1}(\tau/2) - \tau(26 + \tau^2) \sqrt{4 - \tau^2} \right] \right] \right\} + \mathrm{o}\,(1),$$

*as* $d_1 \to \infty$ *while keeping the normalized rank* $\rho$ *constant.*

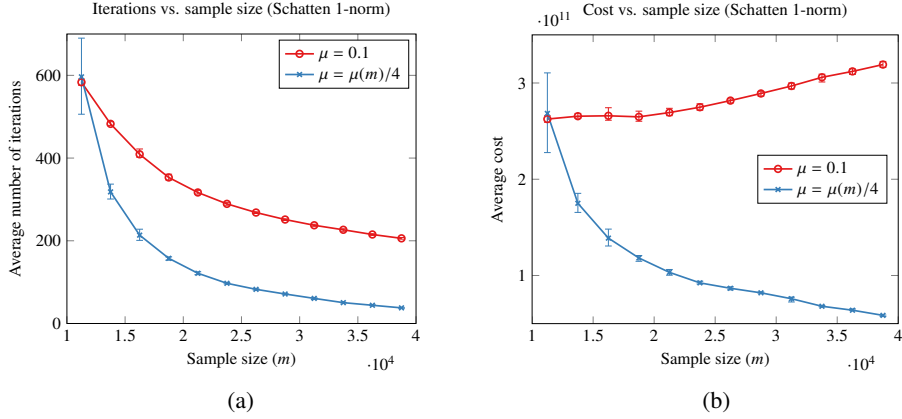

(a)            (b)

Figure 4: **Low-rank matrix recovery experiment.** The average number of iterations (a) and the average cost (b) of 10 random trials of the dual-smoothed low-rank matrix recovery problem with ambient dimension $d = 200 \times 200$ and normalized rank $\rho = 5\%$ for various sample sizes $m$. The red curve represents a fixed smoothing parameter $\mu = 0.1$, while the blue curve uses $\mu = \mu(m)/4$. The error bars indicate the minimum and maximum observed values.

The proof is provided in the supplemental material. The plots of the statistical dimension and maximal smoothing curves closely resemble those of the $\ell_1$ norm and are in the supplemental material as well.

In this case, Algorithm 3.1, line 4 becomes [12, Sec. 4.3]

$$X_z \leftarrow \mu(m)^{-1} \cdot \text{SoftThresholdSingVal}(\text{mat}(A^T z), 1),$$

where mat is the inverse of the vec operator. Given a matrix $X$ with SVD $U \cdot \text{diag}(\sigma) \cdot V^T$,

$$\text{SoftThresholdSingVal}(X, t) = U \cdot \text{diag}(\text{SoftThreshold}(\sigma, t)) \cdot V^T.$$

Algorithm 3.1, line 5 dominates the total cost of each iteration.

## 5.2 The time–data tradeoff

When we adapt the error bound in Proposition 3.1 to this specific problem, the result is nearly same as in the $\ell_1$ case (Proposition 4.2). For completeness, we include the full statement of the result in the supplementary material, along with its proof. Our experience with the sparse vector recovery problem suggests that a tradeoff should exist for the low-rank matrix recovery problem as well.

## 5.3 Numerical experiment

Figure 4 shows the results of a substantially similar numerical experiment to the one performed for sparse vectors. Again, current practice dictates using a smoothing parameter that has no dependence on the sample size $m$ [29]. In our tests, we choose the constant parameter $\mu = 0.1$ recommended by [15]. As before, we compare this with our aggressive smoothing method that selects $\mu = \mu(m)/4$.

In this case, we use the ambient dimension $d = 200 \times 200$ and set the normalized rank $\rho = 5\%$. We test each method with 10 random trials of the low-rank matrix recovery problem for each value of the sample size $m = 11\,250, 13\,750, 16\,250, \ldots, 38\,750$. The measurement matrices are again Gaussian, and the nonzero singular values of the random low-rank matrices $X^\natural$ are 1. We solve each problem with Algorithm 3.1, stopping when the relative error in the Frobenius norm is smaller than $10^{-3}$.

In Figure 4, we see that both methods require fewer iterations for convergence as sample size increases. Our aggressive smoothing method additionally achieves a reduction in total computational cost, while the constant method does not. The observed speedup from exploiting the additional samples is $5.4\times$.

The numerical results show that we have indeed identified a time–data tradeoff via smoothing. While this paper considers only the regularized linear inverse problem, our technique extends to other settings. Our forthcoming work [13] addresses the case of noisy measurements, provides a connection to statistical learning problems, and presents additional examples.

## Footnotes

[1]We compute total cost as $k \cdot md$, where $k$ is the number of iterations taken, and $md$ is the dominant cost of each iteration.

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
