[Supplementary Material]

# Time–Data Tradeoffs by Aggressive Smoothing: Supplementary Material

**John J. Bruer**[1,*]    **Joel A. Tropp**[1]    **Volkan Cevher**[2]    **Stephen R. Becker**[3]

[1]Dept. of Computing + Mathematical Sciences, California Institute of Technology
[2]Laboratory for Information and Inference Systems, EPFL
[3]Dept. of Applied Mathematics, University of Colorado at Boulder
[*]jbruer@cms.caltech.edu

## A   Numerical methodology

This section describes the numerical experiments presented in Sections 4.3 and 5.3. All of the experiments discussed herein were performed on a workstation with a 12-core processor under MATLAB 2014a and OS X 10.9.5.

### A.1   Sparse vector recovery

The data for the sparse vector recovery experiment in Section 4.3 were generated as follows. Fix the ambient dimension $d = 40\,000$. For each value of the sample size $m = 12\,000, 14\,000, 16\,000, \ldots, 38\,000$, perform 10 trials of this procedure, and average the results:

- Generate a sparse vector $x^\natural$ with 2000 nonzero entires placed uniformly at random, each taking the value either $-1$ or $+1$ independently with equal probability.

- Form the random measurement matrix $A \in \mathbb{R}^{m \times d}$ with independent standard Gaussian entires.

- Use the Auslender–Teboulle algorithm (Algorithm 3.1) to solve the dual-smoothed sparse vector recovery problem with both the constant smoothing parameter $\mu = 0.1$ and $\mu = \mu(m)/4$, where $\mu(m)$ is the maximal smoothing parameter (6).

- Stop the algorithm when the primal iterate $x_k$ satisfies $\|x_k - x^\natural\| / \max\{\|x^\natural\|, 1\} < 10^{-3}$.

- Store the number of iterations $k$ and the cost $k \cdot md$ for each choice of the smoothing parameter.

### A.2   Low-rank matrix recovery

The data for the low-rank matrix recovery experiment in Section 5.3 were generated as follows. Fix the ambient dimension $d = d_1 \times d_2 = 200 \times 200 = 40\,000$. For each value of the sample size $m = 11\,250, 13\,750, 16\,250, \ldots, 38\,750$, perform 10 trials of this procedure, and average the results:

- Generate a low-rank matrix $X^\natural := Q_1 Q_2^T$, where the $Q_i$ are chosen uniformly at random from the ensemble of $200 \times 10$ matrices with orthonormal columns (see [1] for the numerical details, as some care must be taken to ensure the appropriate distribution).

- Form the random measurement matrix $A \in \mathbb{R}^{m \times d}$ with independent standard Gaussian entires.

- Use the Auslender–Teboulle algorithm (Algorithm 3.1) to solve the dual-smoothed low-rank matrix recovery problem with both the constant smoothing parameter $\mu = 0.1$ and $\mu = \mu(m)/4$, where $\mu(m)$ is the maximal smoothing parameter (6).

- Stop the algorithm when the primal iterate $X_k$ satisfies $\|X_k - X^{\natural}\|_F / \max\{\|X\|_F, 1\} < 10^{-3}$.
- Store the number of iterations $k$ and the cost $k \cdot md$ for each choice of the smoothing parameter.

**Remark A.1.** We use cost as a proxy for running time for two main reasons. Firstly, the cost gives a representative idea of the amount of work done by the algorithm that matches nicely with the analysis. Secondly, the nature of modern CPUs renders it difficult to accurately compare timing results among many trials. With dynamic frequency scaling, the CPU will run internally at different speeds throughout the run of the experiment in response to surrounding thermal conditions. Typically this means that later trials show higher running time as the CPU has heated considerably over the course of even a moderate-length experiment. This effect is particularly pronounced in laptops.

We did perform separate timing trials, however, that confirm the approximate equivalence between cost and time. Therefore, we feel confident presenting cost as the sole measure of computational work while distancing ourselves from the nuances of CPU frequency scaling.

# B  Statistical dimension calculations

This appendix contains upper bounds on the statistical dimension of the dual-smoothed descent cones presented in the body of the paper. Appendix B.1 examines the dual-smoothed $\ell_1$ norm introduced in Section 4, while Appendix B.2 examines the dual-smoothed Schatten 1-norm introduced in Section 5.

In many practical instances, calculating the statistical dimension of descent cones directly from the definition is infeasible. A polarity argument, originally due to [2], provides an accurate upper bound on the statistical dimension of $\mathcal{D}(f; x)$ depending on the subdifferential $\partial f(x)$.

**Fact B.1** (Statistical dimension of a descent cone [3, Prop. 4.4])**.** *Let $f$ be a proper convex function. Assume that the subdifferential $\partial f(x)$ is compact, nonempty, and does not contain the origin. Then*

$$\delta\left(\mathcal{D}(f; x)\right) \leq \inf_{\tau \geq 0} \mathbb{E}\left[\text{dist}^2\left(g, \tau \cdot \partial f(x)\right)\right].$$

*In addition, the infimum is attained.*

This bound is superb and allows for very accurate approximations of the statistical dimension of descent cones. See [3] for more details. In order to compute the subdifferentials of our dual-smoothed functions, we rely on the following result from subdifferential calculus.

**Fact B.2** (Additivity of subdifferentials [4, Thm. 23.8])**.** *Let $f_1, \ldots, f_m$ be proper convex functions on $\mathbb{R}^d$, and assume that the sets $\text{relint}(\text{dom } f_i)$, for $i = 1, \ldots, m$, share a common point. Then for $f = f_1 + \cdots + f_m$, we have that*

$$\partial f(x) = \partial f_1(x) + \cdots + \partial f_m(x),$$

*for all $x \in \mathbb{R}^d$.*

Given that we dual-smooth a regularizer $f$ by adding a strongly convex function, this fact results in statistical dimension bounds that strongly resemble their unsmoothed counterparts.

## B.1  Dual-smoothed $\ell_1$ norm descent cones

In this section we bound the statistical dimension of the dual-smoothed $\ell_1$ norm by way of the subdifferential bound. We briefly examine how smoothing affects its behavior and provide a simplification for "flat" vectors.

**Proposition B.3** (Descent cones of the dual-smoothed $\ell_1$ norm)**.** *Let $x \in \mathbb{R}^d$ have $s$ nonzero entries. Recall the dual-smoothed $\ell_1$ norm given by*

$$f_\mu(x) := \|x\|_{\ell_1} + \frac{\mu}{2}\|x\|^2.$$

*Then we have the following upper bound on the statistical dimension of its descent cones:*

$$\delta\left(\mathcal{D}(f_\mu; x)\right) \leq \psi(x),$$

*where*

$$\psi(\boldsymbol{x}) := \inf_{\tau \geq 0} \left\{ s\left(1 + \tau^2\right) + 2\mu f_\mu(\boldsymbol{x})\tau^2 + (d - s)\sqrt{\frac{2}{\pi}} \int_\tau^\infty (u - \tau)^2 \, e^{-u^2/2} \, du \right\}.$$

*Proof.* Since the dual-smoothed $\ell_1$ norm $f_\mu$ is invariant under signed coordinate permutations, we assume without loss that $\boldsymbol{x} = (x_1, \ldots, x_s, 0, \ldots, 0)^T$, where the $x_i$ are positive. We will use Fact B.1 to bound the statistical dimension of $\mathcal{D}(f_\mu; \boldsymbol{x})$ in terms of the size of its subdifferentials:

$$\delta\left(\mathcal{D}(f_\mu; \boldsymbol{x})\right) \leq \inf_{\tau \geq 0} \mathbb{E}\left[\operatorname{dist}^2\left(\boldsymbol{g}, \tau \cdot \partial f_\mu(\boldsymbol{x})\right)\right],$$

where $\boldsymbol{g} \sim \text{NORMAL}(\boldsymbol{0}, \mathbf{I}_d)$.

The additivity of subdifferentials, Fact B.2, provides that

$$\partial f_\mu(\boldsymbol{x}) = \partial \|\boldsymbol{x}\|_{\ell_1} + \partial \frac{\mu}{2} \|\boldsymbol{x}\|^2.$$

The function $\boldsymbol{x} \mapsto \frac{\mu}{2} \|\boldsymbol{x}\|^2$ is differentiable with gradient $\mu \boldsymbol{x}$, while the subdiffential of the $\ell_1$ norm has form:

$$\boldsymbol{u} \in \partial \|\boldsymbol{x}\|_{\ell_1} \iff \begin{cases} u_i = 1, & \text{if } i = 1, \ldots, s \\ |u_i| \leq 1, & \text{if } i = s + 1, \ldots, d. \end{cases}$$

Therefore, we find that

$$\boldsymbol{u} \in \partial f_\mu(\boldsymbol{x}) \iff \begin{cases} u_i = 1 + \mu x_i, & \text{if } i = 1, \ldots, s \\ |u_i| \leq 1, & \text{if } i = s + 1, \ldots, d. \end{cases}$$

Since the subdifferential is separable with respect to its coordinates, we compute the distance to the scaled subdifferential coordinatewise as

$$\operatorname{dist}^2\left(\boldsymbol{g}, \tau \cdot \partial f_\mu(\boldsymbol{x})\right) = \sum_{i=1}^{s} \left[g_i - \tau\left(1 + \mu x_i\right)\right] + \sum_{i=s+1}^{d} \max\left\{|g_i| - \tau, 0\right\}^2.$$

Taking the expectation over the Gaussian vector $\boldsymbol{g}$ gives

$$\mathbb{E}\left[\operatorname{dist}^2\left(\boldsymbol{g}, \tau \cdot f_\mu(\boldsymbol{x})\right)\right] = \sum_{i=1}^{s} \left[1 + \tau^2\left(1 + \mu x_i\right)^2\right] + (d - s)\sqrt{\frac{2}{\pi}} \int_\tau^\infty (u - \tau)^2 \, e^{-u^2/2} \, du.$$

We expand and simplify the sum:

$$\mathbb{E}\left[\operatorname{dist}^2\left(\boldsymbol{g}, \tau \cdot f_\mu(\boldsymbol{x})\right)\right] = s(1 + \tau^2) + 2\mu f_\mu(\boldsymbol{x})\tau^2 + (d - s)\sqrt{\frac{2}{\pi}} \int_\tau^\infty (u - \tau)^2 \, e^{-u^2/2} \, du.$$

Taking the infimum over $\tau \geq 0$ completes the proof. $\qquad \square$

We immediately see that the statistical dimension grows monotonically in the smoothing parameter $\mu$ and $\|\boldsymbol{x}\|$. This reinforces the notion that the required number of measurements for exact recovery increases as the problem becomes smoother. In order to calculate the statistical dimension using this result, we need to know the value of $f_\mu(\boldsymbol{x})$, which may not be available. The following corollary allows us to bound the statistical dimension from above as long as we know the scale of the signal.

**Corollary B.4** (Upper bound of the statistical dimension using scale of the signal). *Define the normalized sparsity* $\rho := s/d$. *Then*

$$\frac{1}{d} \delta\left(\mathcal{D}(f_\mu; \boldsymbol{x})\right) \leq \psi(\rho),$$

*where*

$$\psi(\rho) := \inf_{\tau \geq 0} \left\{ \rho \left[1 + \tau^2(1 + \mu \|\boldsymbol{x}\|_{\ell_\infty})^2\right] + (1 - \rho)\sqrt{\frac{2}{\pi}} \int_\tau^\infty (u - \tau)^2 e^{-u^2/2} \, du \right\}.$$

Figure B.1: **Statistical dimension and maximal smoothing for the dual-smoothed Schatten 1-norm.** (a) Upper bounds for the normalized statistical dimension $d_1^{-2}\mathcal{D}(f_\mu; X^\natural)$ of the dual-smoothed low-rank matrix recovery problem for several choices of $\mu$. (b) Lower bounds for the maximal smoothing parameter $\mu(m)$ for several choices of the normalized rank $\rho := r/d_1$.

*Proof.* Note that

$$f_\mu(x) \le s \|x\|_{\ell_\infty} + \frac{s\mu}{2} \|x\|_{\ell_\infty}^2,$$

and the result follows by inserting this into the result of Proposition B.3, rearranging terms, and normalizing by the ambient dimension $d$. □

Clearly, this bound is most accurate when the nonzero values of $x$ are close to $\|x\|_{\ell_\infty}$. As the dynamic range of the signal increases, this bound will increasingly overestimate the statistical dimension. Furthermore, we can adjust $\mu$ to keep the product $\mu \|x\|_{\ell_\infty}$ constant. This means that the smoothing parameter $\mu$ must change with the scale of the signal, but the fundamental geometry—in terms of the statistical dimension—remains the same. As a comparison, we provide the descent cone calculation for the unsmoothed $\ell_1$ norm.

**Fact B.5** (Descent cones of the $\ell_1$ norm [3, Prop. 4.7]). *Let $x \in \mathbb{R}^d$ have $s$ nonzero entries, and define the normalized sparsity $\rho := s/d$. Then*

$$\frac{1}{d}\delta\left(\mathcal{D}(\|\cdot\|_{\ell_1}; x)\right) \le \psi(\rho),$$

*where*

$$\psi(\rho) := \inf_{\tau \ge 0} \left\{ \rho\left(1 + \tau^2\right) + (1 - \rho)\sqrt{\frac{2}{\pi}} \int_\tau^\infty (u - \tau)^2 e^{-u^2/2}\, du \right\}.$$

Note that the unsmoothed problem only depends on the sparsity; changing the scale of the signal or its dynamic range has no effect on the statistical dimension.

## B.2 Dual-smoothed Schatten 1-norm descent cones

Recall that the dual-smoothed Schatten 1-norm of a matrix $X \in \mathbb{R}^{d_1 \times d_2}$ is

$$f_\mu(X) = \|X\|_{S_1} + \frac{\mu}{2} \|X\|_F^2.$$

Now we calculate the statistical dimension of $f_\mu$'s descent cones at low-rank matrices. We will need the following variant of the Marčenko-Pastur Law [5].

**Fact B.6** (Spectral functions of a Gaussian matrix [3, Fact C.1]). *Let $F : \mathbb{R}_+ \to \mathbb{R}$ be a fixed continuous function, and suppose that $p, q \to \infty$ with the ratio $p/q \to y \in (0,1]$, a constant. For $\mathbf{Z}_{pq}$, a $p \times q$ matrix with independent NORMAL$(q, q^{-1})$ entries,*

$$\mathbb{E}\left[\frac{1}{p}\sum_{i=1}^{p} F\left(\sigma_i(\mathbf{Z}_{pq})\right)\right] \to \int_{a_-}^{a_+} F(u) \cdot \varphi_y(u)\,\mathrm{d}u.$$

*The limits of integration $a_\pm := 1 \pm \sqrt{y}$. The kernel $\varphi_y$ is a probability density on $[a_-, a_+]$ given as*

$$\varphi_y(u) := \frac{1}{\pi y u}\sqrt{(u^2 - a_-^2)(a_+^2 - u^2)} \quad \text{for } u \in [a_-, a_+].$$

The following argument closely parallels that of Appendix B.1. In this case, however, we provide a sharp asymptotic result. This creates a clearer connection with the $\ell_1$ case while maintaining a high level of accuracy.

**Proposition B.7** (Descent cones of the dual-smoothed Schatten 1-norm). *Consider a sequence of matrices $[\mathbf{X}(r, d_1, d_2)]$, where $\mathbf{X}(r, d_1, d_2) \in \mathbb{R}^{d_1 \times d_2}$ with $d_1 \le d_2$ and rank $r$. Let $\mathbf{\Sigma}(r, d_1, d_2)$ be the diagonal matrix consisting of the $r$ positive singular values of $\mathbf{x}(r, d_1, d_2)$.*

*Since the dual-smoothed Schatten 1-norm $f_\mu$ is unitarily invariant, we will assume that the first $r$ singular values of $\mathbf{x}(r, d_1, d_2)$ are nonzero for each matrix in the sequence. We let $r, d_1, d_2 \to \infty$ while keeping the following ratios constant:*

$$\rho := r/d_1 \in (0,1)$$
$$\nu := d_1/d_2 \in (0,1]$$
$$\alpha := \frac{\|\mathbf{X}(r, d_1, d_2)\|_{S_1} + \frac{\mu}{2}\|\mathbf{X}(r, d_1, d_2)\|_F^2}{d_1}$$

*Then*

$$\frac{1}{d_1 d_2}\delta\left(\mathcal{D}(f_\mu; \mathbf{X}(r, d_1, d_2))\right) \to \psi(\rho, \nu, \alpha),$$

*where*

$$\psi(\rho, \nu, \alpha) = \inf_{\tau \ge 0}\left\{\rho\nu + (1 - \rho\nu)\left[\rho(1 + \tau^2) + 2\mu\tau^2\alpha + \tau^2\beta \right.\right.$$
$$\left.\left. + (1 - \rho)\int_{a_- \vee \tau(1 - \mu\sigma)}^{a_+} (u - \tau)^2\,\varphi_y(u)\,\mathrm{d}u\right]\right\}.$$

*The limits $a_\pm$ and kernel $\varphi_y$ are as given in Fact B.6, with $y := \nu(1 - \rho)/(1 - \rho\nu)$.*

*Proof.* First, consider a low-rank matrix $\mathbf{X}$ for a fixed $(r, d_1, d_2)$. The subdifferential of the Schatten 1-norm at this matrix takes the form

$$\partial\|\mathbf{X}\|_{S_1} = \left\{\begin{bmatrix} \mathbf{I}_r & \mathbf{0} \\ \mathbf{0} & \mathbf{W} \end{bmatrix} : \sigma_1(\mathbf{W}) \le 1\right\},$$

where $\sigma_1(\mathbf{W})$ is the largest singular value of $\mathbf{W}$ [6, Ex. 2]. By the additivity of subdifferentials, Fact B.2,

$$\partial f_\mu(\mathbf{X}) = \left\{\begin{bmatrix} \mathbf{I}_r & \mathbf{0} \\ \mathbf{0} & \mathbf{W} \end{bmatrix} : \sigma_1(\mathbf{W}) \le 1\right\} + \mu\mathbf{X}.$$

Using the definition of $\mathbf{\Sigma}$, we have that

$$\partial f_\mu(\mathbf{X}) = \left\{\begin{bmatrix} \mathbf{I}_r + \mu\mathbf{\Sigma} & \mathbf{0} \\ \mathbf{0} & \mathbf{W} \end{bmatrix} : \sigma_1(\mathbf{W}) \le 1\right\}$$

To apply the subdifferential bound, Fact B.1, we must calculate

$$\mathbb{E}_G\left[\text{dist}^2\left(\mathbf{G}, \tau \cdot \partial f_\mu(\mathbf{X})\right)\right],$$

where $G$ is a $d_1 \times d_2$ Gaussian matrix. Block-partition $G$ to be compatible with $\partial f_\mu$ to get

$$\text{dist}^2\left(G, \tau \cdot \partial f_\mu(X)\right) = \left\|\begin{bmatrix} G_{11} - \tau(I_r + \mu\Sigma) & G_{12} \\ G_{21} & 0 \end{bmatrix}\right\|_F^2 + \inf_{\sigma_1(W) \leq 1} \|G_{22} - \tau W\|_F^2.$$

By the Hoffman-Wielandt Theorem [7, Cor. 7.3.8],

$$\inf_{\sigma_1(W) \leq 1} \|G_{22} - \tau W\|_F^2 = \inf_{\sigma_1(W) \leq 1} \sum_{i=1}^{d_1-r} [\sigma_i(G_{22}) - \tau\sigma_i(W)]^2$$

$$= \inf_{\sigma_1(W) \leq 1} \sum_{i=1}^{d_1-r} \text{Pos}^2\left(\sigma_i(G_{22}) - \tau\sigma_i(W)\right)$$

$$\leq \sum_{i=1}^{d_1-r} \text{Pos}^2\left(\sigma_i(G_{22}) - \tau\right).$$

We take the expectation and find the first term in the sum as

$$r\left[(d_1 - r) + (d_2 - r) + r + \tau^2\right] + 2\tau^2\mu\left[\|\Sigma\|_{S_1} + \frac{\mu}{2}\|\Sigma\|_F^2\right]$$

This gives a non-asymptotic bound on the statistical dimension:

$$\delta\left(\mathcal{D}(f_\mu; X)\right) \leq \inf_{\tau \geq 0}\left\{r\left[d_1 + d_2 - r + \tau^2\right]\right.$$

$$\left. + 2\tau^2\mu\left[\|\Sigma\|_{S_1} + \frac{\mu}{2}\|\Sigma\|_F^2\right] + \mathbb{E}\left[\sum_{i=1}^{d_1-r}\text{Pos}^2\left(\sigma_i(G_{22}) - \tau\right)\right]\right\}.$$

We recognize the final term in this bound as the expectation of a spectral function of a Gaussian matrix. In order to compute it, we will use the asymptotic result in Fact B.6. First, we normalize our non-asymptotic bound according to the notation given in the statement of the proposition and perform the change of variables $\tau \to \tau\sqrt{d_2 - r}$:

$$\frac{1}{d_1 d_2}\delta\left(\mathcal{D}(f_\mu; X)\right) \leq \inf_{\tau \geq 0}\left\{\rho\nu + (1 - \rho\nu)\left[\rho(1 + \tau^2) + 2\mu\tau^2\alpha\right.\right.$$

$$\left.\left. + (1 - \rho)\mathbb{E}\left[\frac{1}{d_1 - r}\sum_{i=1}^{d_1-r}\text{Pos}^2\left(\sigma_i(\widetilde{G}_{22}) - \tau\right)\right]\right]\right\},$$

where $\widetilde{G}_{22}$ is a rescaled version of $G_{22}$ with independent NORMAL$(0, (d_2 - r)^{-1})$ entries.

Now we apply Fact B.6 with

$$y = \frac{d_1 - r}{d_2 - r} = \frac{\nu(1 - \rho)}{1 - \rho\nu},$$

and obtain

$$\frac{1}{d_1 d_2}\delta\left(\mathcal{D}(f_\mu; X)\right) \to \inf_{\tau \geq 0}\left\{\rho\nu + (1 - \rho\nu)\left[\rho(1 + \tau^2) + 2\mu\tau^2\alpha + (1 - \rho)\int_{a_-\vee\tau}^{a_+}(u - \tau)^2\,\varphi_y(u)\,du\right]\right\}.$$

$$\square$$

In the special case where the matrix $X$ is square, the integral inside the infimum has a closed-form representation.

**Corollary B.8** (Descent cones at square, low-rank matrices)**.** *Let $[X(r, d_1)]$ be a sequence of square matrices, where $X \in \mathbb{R}^{d_1 \times d_1}$ with rank $r$. Let $r, d_1 \to \infty$ while keeping $\rho := r/d_1$ and $\alpha := \left(\|X\|_{S_1} + \frac{\mu}{2}\|X\|_F^2\right)/d_1$ constant. Then*

$$\frac{1}{d_1^2}\delta\left(\mathcal{D}(f_\mu; X(r, d_1))\right) \to \inf_{0 \leq \tau \leq 2}\left\{\rho + (1 - \rho)\left[\rho(1 + \tau^2) + 2\mu\tau^2\alpha\right.\right.$$

$$\left.\left. + \frac{(1 - \rho)}{12\pi}\left[24(1 + \tau^2)\cos^{-1}(\tau/2) - \tau(26 + \tau^2)\sqrt{4 - \tau^2}\right]\right]\right\}.$$

**Corollary B.9** (Upper bound of the statistical dimension using the scale of the signal). *We may bound the statistical dimension result in Corollary B.8 by using the largest singular value $\|X\|$ of the signal.*

$$\frac{1}{d_1^2}\delta\left(\mathcal{D}(f_\mu; X)\right) \le \inf_{0 \le \tau \le 2}\left\{\rho + (1-\rho)\left[\rho\left(1 + \tau^2(1 + \mu\|X\|)^2\right)\right.\right.$$
$$\left.\left. + \frac{(1-\rho)}{12\pi}\left[24(1+\tau^2)\cos^{-1}(\tau/2) - \tau(26 + \tau^2)\sqrt{4-\tau^2}\right]\right]\right\} + o(1).$$

*Proof.* We bound the quantity $\alpha$ as

$$\alpha \le \frac{1}{d_1}\left(r\|X\| + \frac{r\mu}{2}\|X\|^2\right) = \rho\|X\|\left(1 + \frac{\mu}{2}\|X\|\right),$$

and the result follows from rearranging terms. □

Figure B.1(a) shows the statistical dimension curves for a low-rank matrix with singular values equal to 1 for different values of the smoothing parameter $\mu$. As in the $\ell_1$ case, small values of $\mu$ have little effect on the number of measurements required for exact recovery.

We can compare the results of Proposition B.7 with those corresponding to the unsmoothed Schatten 1-norm.

**Fact B.10** (Descent cones of the Schatten 1-norm [3, Prop. 4.8]). *Let $[X(r, d_1, d_2)]$ be a sequence of matrices, where $X \in \mathbb{R}^{d_1 \times d_2}$ with $d_1 \le d_2$ and rank $r$. Suppose that $r, d_1, d_2 \to \infty$ with the limiting ratios $r/d_1 \to \rho$ and $d_1/d_2 \to \nu$. Then,*

$$\frac{1}{d_1 d_2}\delta\left(\mathcal{D}(\|\cdot\|_{S_1}; X(r, d_1, d_2))\right) \to \psi(\rho, \nu),$$

*where*

$$\psi(\rho, \nu) := \inf_{\tau \ge 0}\left\{\rho\nu + (1 - \rho\nu)\left[\rho(1 + \tau^2) + (1-\rho)\int_{a_- \vee \tau}^{a_+}(u - \tau)^2 \varphi_y(u)\,\mathrm{d}u\right]\right\}.$$

*The quantities $y$ and $a_\pm$ along with the integral kernel $\varphi_y$ are as in Fact B.6.*

As in the $\ell_1$ case, we see that the dual-smoothing procedure results in an additional penalty inside the infimum, which scales with the smoothing parameter $\mu$.

# C  Additional proofs

This section contains additional proofs for results in the main text.

## C.1  Proof of Proposition 3.1

This proposition bounds the primal iterates of the Auslender–Teboulle algorithm (Algorithm 3.1). We restate the Proposition 3.1 here.

**Proposition C.1** (Primal convergence of Algorithm 3.1). *Assume that the exact recovery condition holds for the primal problem (3). Algorithm 3.1 applied to the smoothed dual problem (4) converges to an optimal dual point $z_\mu^\star$. Let $x_\mu^\star$ be the corresponding optimal primal point given by (5). Then the sequence of primal iterates $\{x_k\}$ satisfies*

$$\|x^\natural - x_k\| \le \frac{2\|A\|\,\|z_\mu^\star\|}{\mu \cdot k}.$$

*Proof.* This proof uses the same technique that Beck and Teboulle used to prove the equivalent result for their FDPG algorithm in [8]. Recall that we have the Lagrangian

$$\mathcal{L}_\mu(x, z) = f_\mu(x) - \langle z,\ Ax - b\rangle$$
$$= f_\mu(x) - \left\langle A^T z,\ x\right\rangle + \langle z,\ b\rangle.$$

Let $z_k$ be the $k$th iterate of the Auslender–Teboulle algorithm (listed in Algorithm 3.1) for any $k \geq 1$, and let $x_{z_k}$ be the corresponding primal estimate given by

$$x_{z_k} = \arg \min_{x} f_\mu(x) - \left\langle A^T z_k, \, x \right\rangle.$$

Since $f_\mu$ is $\mu$-strongly convex, we have that

$$\mathcal{L}_\mu(x_\mu^\star, z_k) - \mathcal{L}_\mu(x_{z_k}, z_k) \geq \frac{\mu}{2} \|x_\mu^\star - x_{z_k}\|^2.$$

By the definition of the dual function $g_\mu$, we have that $\mathcal{L}_\mu(x_{z_k}, z_k) = g_\mu(z_k)$. Since we assumed that exact recovery holds, we know that $x_\mu^\star = x^\natural$, and so

$$\mathcal{L}_\mu(x_\mu^\star, z_k) = \mathcal{L}_\mu(x^\natural, z_k) = f_\mu(x^\natural) - \left\langle A^T z_k, \, x^\natural \right\rangle + \langle z_k, \, b \rangle$$
$$= f_\mu(x^\natural) = g(z_\mu^\star),$$

where we used strong duality to equate the optimal function values. Therefore,

$$\frac{\mu}{2} \|x^\natural - x_{z_k}\|^2 \leq g(z_\mu^\star) - g(z_k).$$

We can bound the function value of the iterates using the following fact.

**Fact C.2** (Convergence of the Auslender–Teboulle algorithm [9, Thm. 5.2]). *Under the prevailing notation,*

$$g_\mu(z_k) - g_\mu(z_\mu^\star) \leq \frac{2 \|A\|^2 \|z_\mu^\star\|^2}{\mu \cdot k^2},$$

*for any $k \geq 1$.*

Inserting this bound and taking the square root completes the proof. □

## C.2   Proof of Proposition 4.2

This proposition provides an error bound on the primal iterates resulting from the Auslender–Teboulle algorithm (Algorithm 3.1) applied to the dual-smoothed sparse vector recovery problem. We restate Proposition 4.2 from the main text.

**Proposition C.3** (Error bound for dual-smoothed sparse vector recovery). *Let $x^\natural \in \mathbb{R}^d$ with $s$ nonzero entries, $m$ be the sample size, and $\mu(m)$ be the maximal smoothing parameter (6). Given a measurement matrix $A \in \mathbb{R}^{m \times d}$, assume the exact recovery condition (2) holds for the dual-smoothed sparse vector recovery problem. Then the sequence of primal iterates from Algorithm 3.1 satisfies*

$$\|x^\natural - x_k\| \leq \frac{2 d^{\frac{1}{2}} \kappa(A) \left[ \rho \cdot (1 + \mu(m) \|x^\natural\|_{\ell_\infty})^2 + (1 - \rho) \right]^{\frac{1}{2}}}{\mu(m) \cdot k},$$

*where $\rho := s/d$ is the normalized sparsity of $x^\natural$, and $\kappa(A)$ is the condition number of the matrix $A$.*

*Proof.* We apply Proposition C.1 to this problem to obtain that

$$\|x^\natural - x_k\| \leq \frac{2 \|A\| \left\| z_{\mu(m)}^\star \right\|}{\mu(m) \cdot k},$$

where $z_{\mu(m)}^\star$ is the optimal dual point reached by applying Algorithm 3.1 to the dual-smoothed sparse vector recovery problem. In order to adapt the result to this specific problem, we must calculate the size of the optimal dual point $\|z_{\mu(m)}^\star\|$. Since we do not know this point *a priori*, we will instead find an upper bound for *all* optimal dual points.

By the exact recovery assumption and strong duality, the primal estimate in Section 4.1 corresponding to any optimal dual point $z^\star$ is

$$x_{z^\star} = \mu(m)^{-1} \cdot \text{SoftThreshold}(A^T z^\star, 1) = x^\natural.$$

Since the problem is invariant under signed coordinate permutations, assume that the $s$ nonzero values of $x^\natural$ are positive and occupy the first $s$ coordinates. Then,

$$\begin{cases} |\langle a_i, z^\star \rangle|^2 = (1 + \mu(m)x_i^\natural)^2, & \text{if } i = 1, \dots, s \\ |\langle a_i, z^\star \rangle|^2 \leq 1, & \text{if } i = s + 1, \dots, d, \end{cases}$$

where $a_i$ is the $i$th column of $A$. Summing over all columns of $A$ gives

$$\|A^T z^\star\|^2 = \sum_{i=1}^d |\langle a_i, z^\star \rangle|^2 \leq \|\mathbf{e} + \mu(m)x^\natural\|^2,$$

where $\mathbf{e}$ is the vector of all ones.

We can bound the norm $\|A^T z^\star\|^2$ from below by

$$\sigma_{\min}(A)^2 \|z^\star\|^2 \leq \|A^T z^\star\|^2,$$

where $\sigma_{\min}(A)$ is the minimum singular value of $A$. (Recall that $A^T$ is tall.) Therefore,

$$\|z^\star\|^2 \leq \frac{\|\mathbf{e} + \mu(m)x^\natural\|^2}{\sigma_{\min}(A)^2}.$$

In practice, we would not know the true signal $x^\natural$, but it is reasonable to know or have an upper bound on $\|x^\natural\|_{\ell_\infty}$. Using this quantity, we may bound the numerator from above:

$$\|\mathbf{e} + \mu(m)x^\natural\|^2 \leq s(1 + \mu(m) \|x^\natural\|_{\ell_\infty})^2 + (d - s),$$

Taking the square root of the bound for $\|z^\star\|^2$, inserting this into the result of Proposition C.1, and rearranging terms completes the proof. $\square$

## C.3 Error bound for dual-smoothed low-rank matrix recovery

This proposition provides an error bound on the primal iterates resulting from the Auslender–Teboulle algorithm (Algorithm 3.1) applied to the dual-smoothed low-rank matrix recovery problem.

**Proposition C.4** (Error bound for dual-smoothed low-rank matrix recovery)**.** *Let the unknown matrix $X^\natural \in \mathbb{R}^{d_1 \times d_2}$ have rank $r$, $m$ be the sample size, and $\mu(m)$ be the maximal smoothing parameter. Without loss, assume that $d_1 \leq d_2$. Given a measurement matrix $A \in \mathbb{R}^{m \times d_1 d_2}$, assume the exact recovery condition holds for the dual-smoothed low-rank matrix recovery problem. Then the sequence of primal iterates from Algorithm 3.1 satisfies*

$$\|X^\natural - X_k\|_{\mathrm{F}} \leq \frac{2d_1^{\frac{1}{2}} \kappa(A) \left[ \rho \cdot (1 + \mu(m) \|X^\natural\|)^2 + (1 - \rho) \right]^{\frac{1}{2}}}{\mu(m) \cdot k},$$

*where $\rho := r/d_1$ is the normalized rank of $X^\natural$, and $\kappa(A)$ is the condition number of $A$.*

We see the same qualitative behavior of this bound as in the $\ell_1$ case.

*Proof of Proposition C.4.* As in the proof of Proposition 4.2, we must bound the size of all optimal dual points. Let $z^\star$ be any dual optimal point. Recall that

$$\mathrm{SoftThresholdSingVal}(\mathrm{mat}(A^T z^\star), 1) = \arg\min_X \|X\|_{S_1} + \frac{1}{2} \|X - \mathrm{mat}(A^T z^\star)\|_{\mathrm{F}}.$$

Since both the Frobenius norm and Schatten 1-norm are unitarily invariant, we have that

$$\mathrm{SoftThresholdSingVal}(\mathrm{mat}(A^T z^\star), 1) = \mathrm{SoftThresholdSingVal}(\Sigma, 1) = \mathrm{SoftThreshold}(\mathrm{diag}(\Sigma), 1),$$

using the SVD $\mathrm{mat}(A^T z^\star) = U\Sigma V^T$.

Following the proof of Proposition 4.2, we have that

$$\begin{cases} |\Sigma_{ii}|^2 = (1 + \mu(m) \cdot \sigma_i(X^\natural))^2, & \text{if } i = 1, \dots, r \\ |\Sigma_{ii}|^2 \leq 1, & \text{if } i = r + 1, \dots, d_1, \end{cases}$$

where $\sigma_i(X^\natural)$ is the $i$th largest singular value of $X^\natural$. Therefore,

$$\|\mathbf{\Sigma}\|_F^2 = \sum_{i=1}^{d_1} |\mathbf{\Sigma}_{ii}|^2 \le \|\mathbf{I} + \mu(m)\mathbf{\Sigma}_{X^\natural}\|_F^2,$$

where $\mathbf{\Sigma}_{X^\natural}$ is the diagonal matrix containing the singular values of $X^\natural$.

Using the unitary invariance of the Frobenius norm along with a norm equivalence relationship, we have that

$$\sigma_{\min}(A)^2 \|z^\star\|^2 \le \|A^T z^\star\|^2 \le \|A^T z^\star\|_F^2 = \|\mathrm{mat}(A^T z^\star)\|_F^2 = \|\mathbf{\Sigma}\|_F^2.$$

Combining the last two displays and bounding $\|A^T z^\star\|^2$ from above gives that

$$\|z^\star\|^2 \le \frac{\|\mathbf{I} + \mu(m) \cdot \mathbf{\Sigma}_{X^\natural}\|_F^2}{\sigma_{\min}(A)^2},$$

where $\sigma_{\min}(A)$ is the smallest singular value of the (tall) matrix $A$.

Provided that we know (or can bound from above) $\|X^\natural\|$, the maximal singular value of $X^\natural$, we can bound the numerator as

$$\|\mathbf{I} + \mu(m) \cdot \mathbf{\Sigma}_{X^\natural}\|_F^2 \le r \cdot (1 + \mu(m) \|X^\natural\|)^2 + (d_1 - r).$$

Taking the square root of the bound for $\|z^\star\|^2$, inserting this into the result of Proposition C.1, and rearranging terms completes the proof. □