[Reviews · NeurIPS 2014]

Submitted by Assigned_Reviewer_4

This paper pursues an intriguing concept that has been proposed and considered by others as well (as appropriately cited in the present paper): given more data, can we reduce the running time required to obtain a given risk. This is clearly an important question, and while there has been some progress, the surface has really barely been scratched.

This paper considers the problem of sparse regression, in particular in the setting of no noise — the Regularized Linear Inverse Problem. The main idea is to regularize more heavily when there are more data. It has been observed and also proved rigorously, that regularizing increases the rate of convergence, but more interestingly, may not change the solution (by much, or at all). Two important references in this respect are papers by Wotao Yin and Ming-Jun Lai, that demonstrate precisely this idea, with the goal of showing a very similar conclusion as that reached in this paper. In those papers, they show that if the sensing matrix A satisfies some additional properties (e.g., better RIP) then one can regularize and still recover (nearly) the same answer, but without sending the regularizer to zero (just like what is done in this paper). If one has more data, then it is again fairly straightforward calculation to show that the sensing matrix A (assuming it comes from a suitable random ensemble) will indeed satisfy RIP with a stronger constant. Then, since the regularizer controls the strong convexity parameter of the problem, this speeds up convergence (and in particular, guarantees a global linear rate). Another paper that is relevant and has a very similar idea but without using RIP, is by Agarwal, Negahban and Wainwright, where the authors show that thanks to restricted strong convexity and restricted smoothness, gradient methods have global geometric (linear) convergence. The key connection with the present paper is that convergence time depends explicitly on the RSC and RSM parameters, and one can show (as the authors there do) that these improve when one has more data.

The algorithm in the present paper is a dual smoothing algorithm. Strong convexity is exploited in order to convert the dual solution to the corresponding (and unique) primal solution.

The organization and writing of this paper is not as clear as it might be. One thing that would improve the delivery of the results, is some simple calculations for a setting where, say, A comes from Gaussian design. Computing \mu(m) here should not be that difficult, yet would help tell the story. That is, it would be nice to have some calculations analogous to Fact 2.3, but not just for when exact recovery holds, but rather, how big the regularizer can be while still guaranteeing exact recovery. The current connection to regularizing and Fact 2.3 is not completely clear to me; it should be, as it is one of the core pieces of the paper.

Summary: In general the paper is pursuing a nice direction, for which there has been much ground work laid, but the details are not all as clear as they could be. I’m not sure that they are so in this paper either, though the authors are certainly pushing in this direction.

Submitted by Assigned_Reviewer_8

The paper deals with time-data tradeoffs that appear in noiseless (convex) regularized linear inverse problems (RLIP) (e.g. compressed sensing). The authors show that when an excess data is available to solve a RLIP, the regularizer can be smoothed and the RLIP can be relaxed to a simpler problem that can be solved more efficiently in terms of computational complexity. To establish this tradeoff the authors use a standard iterative dual smoothing optimization algorithm and show that in the presence of excess data, minimizing the smoothed regularizer can achieve the same level accuracy with less iterations. On the other hand the authors explicitly characterize when the smoothed RLIP problem will yield the same (true) solution as the original problem by using the notion of the statistical dimension that precisely characterizes the phase transition curve of RLIPs with random Gaussian sensing matrices. The authors present their results for the problems of sparse signal estimation via the l-1 norm, and low rank matrix estimation via the nuclear norm. They also point to other examples and noisy signal estimation in subsequent work.

Quality: The authors use modern convex analysis tools (e.g. statistical dimension of convex cones) to approach the problem of time-data tradeoffs in statistical estimation in a principled and well structured way.
Clarity: The paper is written in a very clear way and is self-contained.
Originality and significance: The authors cite the innovative work of Chandrasekaran and Jordan as another example of time-data tradeoffs in RLIP. However, they propose a different approach (smoothing the regularizer instead of enlarging the constraint set) which appears to be more flexible and applicable to problems that appear often in practice.

My only concern is the dependance of the method on the choice of the optimization algorithm. Are these results also applicable to other more specialized algorithms (e.g. approximate message passing for l-1 minimization) that offer strong performance guarantees?
Summary: The paper addresses the problem of time data tradeoffs in regularized linear inverse problems in a novel way, that is theoretically sound, and practically applicable.

Submitted by Assigned_Reviewer_36

In regularized linear inverse problems, the sample complexity for recovering the true signal has recently been well understood. This paper agues that excess observations than needed in the sample complexity bound can be used to reduce the computational cost. This is achieved through carefully smoothing the original nonsmooth regularizer (such as the ell_1 norm) hence reducing computational iterations, and without violating the sample complexity bound for recovery. The paper is clearly written and the idea seems interesting, however, there are several practical issues that perhaps should have been addressed.

The reviewer found the idea to squeeze more smoothing from excess observations quite interesting, however, this is at the expense of potentially compromising the recovery probability (eta in Fact 2.3, in particular the success recovery bound). A bit disappointingly, in the experiments the authors did not simulate the probability of "exact" recovery at all; instead, only some predefined "closedness" to the true signal is used to declare victory of the algorithm. The authors did decrease the smoothing parameter by a factor of 4 to hedge against failure, but this is adhoc and insufficient. It would be very interesting (in some sense also necessary) to see whether or not, or to what extent, does the aggressive smoothing affect the (exact) recovery probability. Judging from the bound (Fact 2.3), there should be another tradeoff.

The experiments can be improved. The reviewer expected to see the following comparison: When we compare against some constant smoothing parameter, say mu = 0.1, we can compute the number of samples needed for high probability recovery, call it m_mu, which is smaller than the number of available samples m. Then we just randomly throw away m - m_mu samples. This way one also reduces the computational cost by reducing the size of the problem. In the reviewer's opinion, this should be the "conventional" constant smoothing algorithm to compare to. Right now, the authors seem to let the competitor run on all available samples, putting it in a disadvantageous position. The two strategies (throwing away redundant data or aggressively smoothing) are really two blades of the same sword: both aim at maintaining the recovery bound in the minimal sense. But the first approach is even more appealing: it certainly requires less memory. The reviewer strongly recommends the authors to perform a serious comparison and report the relative strengths and weaknesses.

Another issue the reviewer would like to see addressed is practicality: so far the analysis and experiments are done with prior knowledge of the true signal. In practice this is not available. How could one still be able to apply the aggressive smoothing without being too aggressive? On the other hand, it all appears to "tune" the smoothing parameter, which is what is done in practice anyways. From a fully practical aspect, what is new and different here then? If the new message (as argued in line 325) is that the smoothing parameter needs to depend on the sample size, then how can this be implemented without too much prior knowledge of the true signal?
Summary: The paper did a good job in explaining and designing a strategy on how excess samples in regularized linear inverse problems can be exploited computationally to speed up convergence. The paper is very well-written and the idea is interesting. However, there are several critical issues both theoretically and experimentally that perhaps should have been addressed.
Author Feedback
Author rebuttal: We argue that we can reduce the computational cost of statistical optimization problems by aggressively smoothing when we have a large amount of data. While this idea seems very natural in retrospect, we believe that it is a new and important principle. The case study presented provides theoretical and experimental evidence that such a time–data tradeoff exists in one instance. We believe there are many other examples to explore, and we have identified several beyond the scope of the current work. In order to address the concerns raised during the review process, we submit our answers to the three questions posed by the meta-reviewer.

Question 1: One reviewer mentions two papers that he or she believes reach very similar conclusions to our own. We will address these works and their relationship to our work specifically. We believe, however, that our conclusion is new and different in character from the previous works. In particular, neither paper identifies a time–data tradeoff, nor can you derive one directly from their results.

Lai and Yin indeed consider the sparse vector and low-rank matrix problems that we do in our case study. They propose choosing the smoothing parameter based on the l∞ norm of the signal (in the sparse vector case) and the largest singular value of the signal (in the low-rank matrix case). We, however, calculate the location of the phase transition for exact recovery in the dual-smoothed problem and choose a smoothing parameter based on the sample size as well. Our method results in a greater amount of smoothing and better performance as sample size increases. This is the time–data tradeoff.

Agarwal, Negahban and Wainwright show that some nonsmooth optimization problems in statistics have global linear convergence rates due to the properties of restricted strong convexity (RSC) and restricted smoothness (RSM). They also show that the RSC/RSM parameters improve with sample size. They do not, however, discuss a time–data tradeoff explicitly. And while they show that iteration count decreases as sample size increases, the overall computational cost rises (as evidenced by the constant smoothing scheme in Figures 3 and 4 of our work). That is, the properties of RSC/RSM alone do not lead to a time–data tradeoff; the aggressive smoothing we impose does.

Question 2: One reviewer proposed that, given a fixed smoothing parameter, we should throw away excess samples in order to speed up the computation. In our framework, this is equivalent to simply reducing the sample size from the outset. For the sparse vector case, the smallest sample size we test is m = 300. Since we cannot reduce m greatly in this case while guaranteeing exact recovery, we feel justified in using m = 300 as a baseline value. Our result in Figure 3(b) clearly shows that our aggressive smoothing scheme, at every sample size m tested, has cost lower than the constant smoothing scheme at the baseline m = 300. The same applies for the low-rank matrix experiment (Figure 4(b)). That is, throwing away excess samples under the constant smoothing scheme still results in worse performance than our aggressive smoothing scheme.

The reviewer is indeed correct that the best course of action under the constant smoothing scheme would be to throw away data. Our principle, however, is to exploit excess samples to reduce computational time via aggressive smoothing. The results of our experiments highlight the benefit of doing so (and the cost of not doing so).

Question 3: The case study we present relies on the Gaussian measurement assumption when calculating the location of the phase transition. There is evidence in the literature that problems of this type exhibit some universality, and so it is reasonable to expect that other measurement matrices would work in practice.

One reviewer questions whether or not it is practical to compute the smoothing parameter without too much prior knowledge of the signal. A calculation of this parameter requires knowledge of the sparsity (or rank) and magnitude of the signal, but an estimate will suffice. A conservative estimate will decrease the effectiveness of our method (by resulting in a less aggressive choice of the smoothing parameter μ(m) as a function of the sample size m), but it will still adapt the smoothing to the sample size and perform better than smoothing using a small, constant parameter. Furthermore, exact recovery will still hold.

Our principle of aggressive smoothing is practical, and it is not equivalent to adapting the smoothing parameter in an ad hoc manner to any given data set.